# Cancer Patients’ Satisfaction with In-Home Palliative Care and Its Impact on Disease Symptoms

**DOI:** 10.3390/healthcare11091272

**Published:** 2023-04-29

**Authors:** Inmaculada Valero-Cantero, Cristina Casals, Milagrosa Espinar-Toledo, Francisco Javier Barón-López, Francisco Javier Martínez-Valero, María Ángeles Vázquez-Sánchez

**Affiliations:** 1Puerta Blanca Clinical Management Unit, Malaga-Guadalhorce Health District, 29004 Malaga, Spain; 2ExPhy Research Group, Department of Physical Education, Instituto de Investigación e Innovación Biomédica de Cádiz (INiBICA), Universidad de Cádiz, 11519 Puerto Real, Spain; 3Rincón de la Victoria Clinical Management Unit, Malaga-Guadalhorce Health District, 29730 Malaga, Spain; 4Faculty of Health Sciences, Institute of Biomedical Research in Málaga (IBIMA), University of Malaga, 29016 Malaga, Spain; 5Midlothian Foot Care, Dalkeith and National Health Service, Dalkeith EH22 1DU, UK; 6Department of Nursing, Faculty of Health Sciences, PASOS Research Group and UMA REDIAS Network of Law and Artificial Intelligence Applied to Health and Biotechnology, University of Malaga, 29071 Malaga, Spain

**Keywords:** medical oncology, quality of life, patient satisfaction, symptom assessment, palliative care, home care services, community health nursing

## Abstract

The aim of the study was to determine whether the satisfaction of cancer patients with in-home palliative care is associated with the impact of disease symptoms and with self-perceived quality of life. This was a cross-sectional descriptive study, conducted in the primary health care sector in six clinical management units, where 72 patients were recruited over a period of six months. The severity of symptoms was determined by the Edmonton Symptom Assessment System (ESAS). Quality of life was evaluated with the EORTC QLQ-C30 (version 3) questionnaire, and patients’ satisfaction with the care received was evaluated by the Client Satisfaction Questionnaire (CSQ-8). The patients’ satisfaction with the health care received was represented by an average score of 6, on a scale of 1–10; thus, there is room for improvement in patient satisfaction. Moreover, it was found that more intense symptoms and lower quality of life are associated with lower satisfaction with health care received (*p* = 0.001). Similarly, when symptoms are more severe, the quality of life is lower (*p* < 0.001). The identification of fatigue, reduced well-being, pain, drowsiness, and depression as the symptoms experienced with the highest intensity by our patients provides valuable information for health care providers in developing individualized symptom management plans for patients with advanced cancer.

## 1. Introduction

Worldwide, cancer is a major cause of mortality, responsible for one in every six deaths, according to the World Health Organization [1]. Patients with advanced cancer need palliative care as early as possible [2,3], since they present changeable, severe symptoms that require continuous, high-quality health care [4]. This highlights the need for increased awareness and resources to be dedicated to palliative care for cancer patients.

Among other symptoms, patients may experience pain, fatigue, nausea, depression, anxiety, drowsiness, dyspnoea, sleep disorders, loss of appetite, constipation, and altered quality of life [5,6,7,8], and these symptoms usually worsen as the disease progresses [9]. Palliative care is aimed at relieving these symptoms and improving the patient’s quality of life, as well as providing support for their families and caregivers, by addressing their physical, psychological, social, and spiritual needs.

In order to provide effective symptom management and improve the quality of life for patients with advanced cancer, healthcare providers must have reliable and accurate measures of the symptoms they are experiencing. The mentioned symptoms can be measured in various ways, but one of the most widely used instruments, both in clinical practice and in research, is the Edmonton Symptom Assessment Scale (ESAS) [10,11,12], which evaluates the intensity of ten physical and psychological symptoms. A major advantage of this scale is its ease of use [13]. Given the significant impact that advanced cancer and its related symptoms have on patients’ well-being, it is essential that healthcare providers prioritize the regular assessment of symptoms in their care management plans.

One of the fundamental objectives of health care for these patients is to improve their quality of life, which is inevitably diminished by the oncological process and the symptoms produced [14]. Quality of life is a multidimensional construct that encompasses various aspects of an individual’s life, including physical health, mental health, social functioning, and general well-being [15]. It is a subjective measure that reflects an individual’s perception of their overall life satisfaction. Thus, assessing the patient’s quality of life should be also an integral part of their overall health care management.

Quality health care should alleviate the symptoms (or at least maintain the status quo, or slow the worsening) presented by a cancer patient in palliative care, and hence improve the quality of life. However, to decide whether this care is really effective, the patient’s own assessment of outcomes must be obtained, and this aspect of the question has received relatively little research attention [16]. Patient-reported outcomes have become increasingly important in assessing the effectiveness of health care interventions and should be considered in the evaluation of palliative care for cancer patients.

A further consideration is the fact that the patient’s preferences in health care planning should be taken into account in the clinical setting. In this respect, some recommendations have already been offered [17]. The recommendations involve tailoring advance care planning to the readiness of the individual, adjusting the content of advance care planning as the individual’s health condition worsens, and utilizing trained non-physician facilitators to support the advance care planning process [17]. Patients’ satisfaction with the health care received is of fundamental importance and is closely related to the degree of concordance between the actual treatment and care received and the patient’s preferences in this regard [18], and this consideration is as valid for patients receiving in-home palliative care as for those who are hospitalized [19].

On the other hand, according to prior research in the area of palliative care, advance health care planning between physicians and patients does not increase the latter’s satisfaction with the medical care received [20], nor does it produce differences in treatment, in perceived quality of life, or in physical and mental symptoms [21]. In consequence, joint planning of the health care program between physicians and patients may not be strictly necessary. Nevertheless, an assessment of the patient’s continuing acceptance of the treatment and care received may be useful, and this question can be assessed by means of user satisfaction surveys.

Our study aim is to determine whether satisfaction with the health care received is associated with the severity of symptoms and with self-perceived quality of life for patients with advanced cancer receiving in-home palliative care. The results of this study can be used to inform health care providers on how to improve care for advanced cancer patients receiving in-home palliative care, with the aim of improving patients’ satisfaction, reducing the severity of symptoms, and enhancing their quality of life.

## 2. Materials and Methods

### 2.1. Design

The study employed a multicenter, cross-sectional descriptive design. It was conducted in the field of primary health care in six clinical management units in the Málaga-Guadalhorce Health District (Málaga, Andalusia, Spain).

### 2.2. Eligibility Criteria and Sampling

The following inclusion criteria were applied: (1) Cancer patients receiving in-home palliative care, (2) who were aged 18 years or older. The exclusion criteria were: (1) Patients with highly advanced disease, resulting in a life expectancy of only a few days, or (2) patients with advanced stage dementia or psychological disorders that would impair their ability to make rational decisions. All individuals included in the study had previously received treatment for cancer through surgery, radiotherapy, or chemotherapy. At the time of the study, none of the patients were receiving systemic treatment, and the treatment provided was focused on symptom improvement.

The study sample was constituted from the cancer patients who according to the corresponding Digital Clinical History (DIRAYA, Spanish initials) were currently receiving palliative care. DIRAYA, defined as an integrated management and information system for health care, is the system used in the Andalusian Health Service as a support for electronic medical records.

### 2.3. Measures

Several measures were utilized to collect data from participants, which were carefully selected based on their validity and reliability. The measures used in the present study are detailed as presented below.

#### 2.3.1. Sociodemographic Data and Clinical Characteristics

The sociodemographic information included: age, sex, marital status, and education. The clinical characteristics included: type of cancer and duration of palliative care. This information was recorded to describe the study sample and consider the generalizability of the findings.

#### 2.3.2. Severity of Disease Symptoms

Assessment of symptoms was conducted using the Edmonton Symptom Assessment System (ESAS) scale [10,22]. ESAS has been psychometrically validated and translated into over 20 languages with good internal reliability (Cronbach α 0.79), test-retest reliability (Spearman correlation coefficient 0.86 on Day 2 and 0.45 on Day 7), and convergent validity [10]. Moreover, the ESAS is a valid, reliable, responsive, and feasible instrument with adequate psychometric properties when tested on Spanish advanced cancer patients [23].

This instrument measures the average intensity of ten common symptoms experienced by cancer patients during the previous week. These symptoms include pain, fatigue, nausea, depression, anxiety, drowsiness, dyspnea, loss of appetite, reduced well-being, and sleep disorders. Patients rated the severity of each symptom at the time of evaluation on a scale from 0 to 10, where 0 indicates the symptom was absent and 10 that it was of the worst possible severity. Based on the ESAS symptoms, following outcomes were obtained:
-Physical ESAS score: calculated as the sum of pain, fatigue, nausea, drowsiness, appetite, and dyspnea symptoms (ranging from 0 to 60).-Emotional ESAS score: calculated as the sum of depression and anxiety symptoms (ranging from 0 to 20).-The total ESAS score: calculated as the sum of all ten symptoms (ranging from 0 to 100).

#### 2.3.3. Quality of Life

The European Organization for Research and Treatment of Cancer Quality of Life Questionnaire-Core 30 (EORTC QLQ-C30) score was developed to assess the quality of life of cancer patients. The EORTC QLQ-C30 questionnaire version 3.0 of this instrument includes 30 questions covering five functional domains (physical, role, cognitive, emotional, and social), eight symptoms (fatigue, nausea/vomiting, pain, dyspnoea, insomnia, loss of appetite, constipation, and diarrhea), and the financial impact produced by the disease. The questionnaire is scored on a four-point scale (ranging from 1 = “Not at all” to 4 = “A lot”), and these questions are scored from 0 to 100. In addition, general health and quality of life are rated on a seven-point scale ranging from 1 (=“Very poor”) to 7 (=“Excellent”), with a score range of 0 to 100 representing the patient’s overall health status, where 100 is the best possible condition [24]. High scores on the global health and functional scales indicate better quality of life, whereas on the symptom scale it would indicate decreased quality of life as it indicates the presence of cancer-related symptoms [24].

The validity and reliability of the Spanish version of the EORTC QLQ-C30 have been demonstrated as an effective tool for assessing quality of life in cancer patients. The reliability of the Spanish version of the questionnaire was found to be greater than 0.86, and the total score of the scale was a good indicator of the quality of life of cancer patients [25].

#### 2.3.4. Satisfaction with the Health Care Received

This parameter was addressed using the Client Satisfaction Questionnaire-8 (CSQ-8), which includes eight Likert-scale questions rated from 1 to 4, resulting in a total score ranging from 8 to 32 points with higher scores indicating more satisfaction with the health care. Additionally, the questionnaire includes three open-ended questions asking patients to identify what they liked most and least about their health care, and what changes they would suggest [26].

The CSQ-8 is a brief, global index rating scale reliable in a variety of service settings to measure satisfaction with the health care received. Internal consistency reliability ranged from 0.83 to 0.90, supporting that the Spanish version of the questionnaire was reliable, valid, and linguistically equivalent to the English version [27]. 

### 2.4. Ethical Issues and Permissions

The study was granted ethical approval by the Malaga Provincial Ethics Committee, with project code 1752-N-18. All patients were informed about the study aims and methods both verbally and in writing. Signed informed consent to participate was requested and obtained from all participants. This project received funding support from the Regional Ministry for Health and Families in the field of primary care, through the Andalusian Health Service (SAS), under the project code AP-0157-2018. We took measures to protect the privacy and confidentiality of our participants, including the use of anonymized data and secure storage of personal information. We also disclosed any potential conflicts of interest, such as relationships with funding sources or commercial entities involved in the study, to ensure transparency and integrity in our research practices.

### 2.5. Data Collection

The patients were recruited to the study over a six-month period from January to December 2022. The final study sample was composed of 72 cancer patients receiving in-home palliative care, who met all the criteria for inclusion and who accepted the invitation to take part. A nurse subsequently visited these patients in their homes and asked them to complete the corresponding questionnaires.

### 2.6. Statistical Methods

The statistical data obtained are presented as mean ± standard deviation (SD) for the quantitative variables, and as absolute frequency (n) and percentage (%) for the categorical ones. Bivariate associations between the CSQ-8, EORTC QLQ-C30 and ESAS scores were analyzed using Spearman’s nonparametric correlation test. Moreover, multiple linear regression was applied with the CSQ-8 as the predicted outcome and with EORTC QLQ-C30 and ESAS as independent variables. The SPSS 22 statistical software was used for all these analyses. Statistical significance was considered at a *p*-value of less than 0.05.

## 3. Results

### 3.1. Recruitment

Of the 78 patients initially assessed, three were excluded due to not meeting the eligibility criteria, two declined the invitation, and one passed away before the scheduled interview. The sociodemographic and clinical characteristics of the final sample (n = 72) are presented in Table 1.

### 3.2. Study Variables

In our sample, the evaluation of the patients’ symptoms is presented in Table 2. The mean of the total ESAS score of cancer patients was 32.25 ± 15.69 points, with a range from 0 to 100 points, where 0 is the best possible condition.

The perceived quality of life, based on the EORTC QLQ-C30 findings, is described in Table 3. The mean of the Global Health Status of cancer patients was 46.30 ± 23.27 points, with a range from 0 to 100 points, where 100 is the best possible condition.

Finally, descriptive results regarding the patients’ satisfaction with health care assessed through the CSQ-8 are shown in Table 4. The mean total satisfaction score was 19.72 ± 3.34 points, with a range from 8 to 32 points, where 32 is the best possible condition.

### 3.3. Bivariate Associations and Regression Analysis

The bivariate relationships between the EORTC QLQ-C30 (Global Health Status), ESAS scale score, and CSQ-8 are presented in Table 5, and significant associations were found in all cases.

The bivariate associations of the CSQ-8 with the level of ESAS symptoms showed that a higher satisfaction with the health care was significantly associated with lower severity of symptoms experienced by cancer patients (Spearman’s rho = −0.397, *p* = 0.001).

The bivariate associations of the CSQ-8 with the Global Health Status assessed with the EORTC QLQ-C30 reported that the patients who presented higher satisfaction with their health care also demonstrated significantly higher levels of perceived Global Health Status (r = 0.486, *p* < 0.001).

Moreover, a higher quality of life (the EORTC QLQ-C30 total score) was associated with lower severity of symptoms assessed with the ESAS scale (r = −0.511, *p* < 0.001).

Using the CSQ-8 scale as the dependent variable, multiple linear regression analysis is presented in Table 6. An increment of 1 point on ESAS was related to a 0.5-point decrease of CSQ-8, while an increment of 1 point on EORTC QLQ-C30 was related to a 0.64-point increase in CSQ-8.

## 4. Discussion

The study sample comprised of an equal proportion of men and women with an average age of 71 years. The most commonly observed types of cancer were colon, breast, and lung cancer, with frequencies of occurrence comparable to those reported in international studies [28]. The fact that the most commonly observed types of cancer in this study were consistent with those reported in international studies suggests that the study findings may be applicable to a broad range of patients with advanced cancer.

The mean duration of palliative treatment in our study can be considered acceptable, but earlier referral would be preferable. However, other studies have reported even shorter durations of palliative care, from referral until death [29]. It is now generally agreed that therapeutic and palliative care for cancer patients should be more closely integrated, as this could enhance the overall care received [2].

The ESAS results revealed that fatigue, reduced well-being, pain, drowsiness, and depression were the symptoms experienced with the highest intensity. These symptoms were previously identified in another study [30], although at a slightly higher severity level. In our sample, these symptoms were classified as having moderate severity (scores ranging from 4 to 6); nevertheless, the individual experience of each patient may vary [31,32]. A major barrier to adequate symptom treatment is poor assessment [33]. Symptom assessment is initiated with the use of standardized scales emphasizing anxiety, depression, physical symptoms, and coping strategies. These scales help to assess the severity of symptoms and guide symptom management. For effective palliative care, it is essential to use a standardized approach to symptom assessment that enables accurate identification of symptoms and their impact on patient’s quality of life [5].

The average score for overall quality of life, as measured by the Global Health Status findings, among these patients with advanced cancer was >47 points, indicating a moderate-to-poor level of quality of life. This value is consistent with that found in comparable studies [34,35]. The finding that patients with advanced cancer in this study reported a moderate-to-poor level of quality of life highlights the importance of providing effective supportive care interventions to address symptoms and improve overall well-being. Given that this result is consistent with findings from comparable studies, it suggests that health care providers should prioritize interventions that have been shown to be effective in improving quality of life in this patient population. Additionally, the use of the Global Health Status measure provides a standardized tool for assessing quality of life in patients with advanced cancer, which can be useful for clinical decision-making and evaluating the effectiveness of interventions over time. Overall, these findings can inform the development and implementation of interventions aimed at improving quality of life for patients with advanced cancer.

On a scale of 8 to 32 points (where 32 represents the best possible condition), the patients’ satisfaction with the healthcare they received was around 20 points. The item on the CSQ-8 survey that received the lowest score was related to “How satisfied are you with the amount of help you received?”, highlighting the need for greater support for in-home palliative care. Out of all the items on the survey, only two (“How would you rate the quality of service you received?” and “In an overall, general sense, how satisfied are you with the service you received?”) received a mean score higher than 3 points on a scale of 1 to 4. These findings suggest that there is room for improvement in patient satisfaction, as well as in the severity of the symptoms they experienced and their overall quality of life. The evaluation of satisfaction with care is a valuable approach for assessing the quality of cancer care. Satisfaction with care represents a significant domain in cancer outcomes research, although its full potential has yet to be explored [36].

Regarding the bivariate associations observed, it was found that more intense symptoms and lower quality of life are associated with lower satisfaction with health care received [36]. Similarly, when symptoms are more severe, the quality of life is lower. Conversely, patients who are most satisfied with the health care received usually experience less severe symptoms and have a better quality of life. The regression analysis showed that an increase of one point on the symptoms scale (assessed with ESAS) was associated with a decrease of half a point on the satisfaction scale (assessed with CSQ-8). Similarly, an increase of one point on the scale of quality of life (assessed with EORTC QLQ-C30) was associated with a slightly greater than half a point increase on the satisfaction scale.

These findings suggest that symptom control and improvement in quality of life are important factors in enhancing patient satisfaction with health care. While the relationship between patients’ satisfaction with health care, quality of life, and symptom intensity may seem obvious, it is important to empirically demonstrate and quantify these associations to emphasize the significance of understanding patients’ experiences in this regard.

A previous study [37] reported that, compared to usual oncology care, a concurrent nurse-led, palliative care-focused intervention addressing physical, psychosocial, and care coordination led to higher scores for quality of life and mood, but not to improvements in symptom intensity scores or reduced hospitalization or emergency department visits. Thus, the early integration of palliative care has been shown to enhance the quality of life and satisfaction with care and is now more widely recommended for patients with advanced cancer [38]. Providing early palliative care enhances satisfaction with care in advanced cancer by effectively addressing patients’ emotional distress and quality of life, improving collaborative relationships with healthcare providers, and addressing concerns about end-of-life preparation [39]. Moreover, a recent study concluded that an early integration of palliative care is recommended for patients with advanced cancer, as it has been shown to both enhance quality of life and alleviate symptom burden [40].

Notwithstanding, a Cochrane review [41] suggested that there is low-quality evidence supporting that compared to usual care, specialist palliative care may provide small benefits for patient outcomes such as health-related quality of life, symptom burden, and patient satisfaction with care. Therefore, more well-conducted studies are needed to draw stronger conclusions and to assess the cost-effectiveness of this palliative care [41]. Terminal cancer patients require extensive and continuous care from health care professionals, but the causes of dissatisfaction and ways to improve it are not clear from the literature review [42], since it requires a comprehensive assessment of patients’ satisfaction with care [43,44].

The present study finds that patients’ dissatisfaction is associated with decreased quality of life and increased multiple symptoms among advanced cancer patients. This highlights the need for improved care for this patient population. Identifying the factors that contribute to the dissatisfaction of patients with terminal cancer can lead to the development of interventions and strategies that improve their quality of life and overall satisfaction with care. For instance, health care providers could focus on managing symptoms more effectively, improving communication and information-sharing with patients, and involving patients and their families in shared decision-making. Such improvements could ultimately lead to better patient outcomes and experiences.

## 5. Limitations

The present study is subject to the following limitations, and the results should be interpreted accordingly. Firstly, it is cross-sectional, and so the direction and causality of the effects recorded cannot be inferred. Additionally, information on the duration of time since the suspension of systemic treatment was not collected, which may have influenced the observed outcomes. Moreover, variables other than those studied may also influence patients’ satisfaction with their health care. Hence, randomized controlled trials are encouraged to analyze the effect of interventions aimed at improving satisfaction with care and their relationship with quality of life and symptom severity in palliative care patients with cancer.

Furthermore, the sample size was limited, due to the relatively small number of cancer patients in our population receiving in-home palliative care. Although the present study is multicenter, involving six health centers in Andalusia, Spain, it may not be fully representative of the entire population of cancer patients receiving palliative care in other regions or countries. Therefore, the generalizability of the study findings may be limited. Additionally, cultural and socioeconomic differences between regions or countries may also influence patients’ experiences and satisfaction with health care, which should be taken into consideration when interpreting the results of this study. Moreover, the study may be affected by selection bias, as patients who agreed to participate in the study may differ from those who declined or were unable to participate. Finally, self-reported symptoms may be subject to recall bias or individual interpretation.

## 6. Conclusions

The study shows that dissatisfaction among advanced cancer patients is linked to lower quality of life and increased symptoms, emphasizing the need for better care. Indeed, patients’ satisfaction with the healthcare they received was around 20 points out of 32, indicating room for improvement in patient satisfaction and the severity of symptoms. The evaluation of satisfaction with care is important in assessing the quality of cancer care, and there is a need to explore its full potential for in-home palliative care.

The study found that fatigue, reduced well-being, pain, drowsiness, and depression were the most intense symptoms experienced by patients with moderate severity. Standardized scales are important in assessing the severity of symptoms and guiding symptom management and care. The use of the ESAS tool provides a standardized way to assess symptom intensity, which can aid in identifying patients who may require additional interventions and monitoring the effectiveness of interventions over time.

Moreover, the study’s finding that patients with advanced cancer had a moderate-to-poor quality of life underscores the need for effective supportive care interventions. Given the consistency with similar studies, healthcare providers should prioritize proven interventions. The use of the standardized Global Health Status measure is useful for evaluating the effectiveness of interventions and guiding clinical decisions. These findings can inform the development of interventions aimed at improving quality of life for patients with advanced cancer.

When in-home palliative care is provided to patients with advanced cancer, it is important to determine their satisfaction with this process, together with their assessment of the symptoms presented. Both factors are relevant to the quality of health care and, ultimately, to the self-perceived quality of life. Identifying factors that contribute to dissatisfaction can lead to interventions that improve patient satisfaction, such as improving symptom management, communication with patients, and involving them in decision-making. Such improvements could lead to better patient outcomes and experiences.

In conclusion, these findings can be applied in practice by encouraging health care providers to routinely assess and address patient satisfaction during in-home palliative care. This may help to alleviate symptoms and improve quality of life for patients, while also providing them with emotional and social support during a difficult time. In addition, early referral to palliative care services can help to ensure that patients receive appropriate end-of-life care, such as advance care planning and symptom management, in a timely and compassionate manner. While the mean duration of palliative treatment in our study was considered acceptable, efforts to promote earlier referral to palliative care services may further improve the quality of care received by patients with advanced cancer.

Further research is necessary to identify effective interventions and establish a causal relationship between satisfaction with care and improved patient outcomes in the context of palliative care for cancer patients. These studies will help inform the development of evidence-based practices and policies for providing optimal palliative care to cancer patients in need.

## Figures and Tables

**Table 1 healthcare-11-01272-t001:** Sociodemographic and clinical characteristics of palliative cancer patients (n = 72).

**Sociodemographic Variables**	**M**	**SD**
Age (years)	74.61	10.13
	**N**	**%**
Gender		
Male	39	54.2
Female	33	45.8
Marital status		
Married	38	50.8
Widowed	23	31.9
Single	9	12.5
Divorced	2	2.8
Education level		
Primary education	29	40.3
No formal education	21	29.2
Secondary education	16	22.2
Higher education	6	8.3
**Clinical variables**	**M**	**SD**
Duration of palliative care (months)	4.88	5.84
	**N**	**%**
Type of cancer		
Colon	13	18.1
Lung	10	13.9
Breast	7	9.7
Pancreatic	6	8.3
Rectal	5	6.9
Prostate	4	5.6
Liver	4	5.6
Oropharyngeal	4	5.6
Kidney	4	5.6
Lymphoma	4	5.6
Bladder	3	4.2
Brain	3	4.2
Cervical	3	4.2
Ovarian	2	2.8

M: Mean, SD: Standard Deviation.

**Table 2 healthcare-11-01272-t002:** Results of the Edmonton Symptom Assessment System of 72 palliative cancer patients.

**Symptoms** **(Range 0 to 10, 10 Being the Worst Possible Severity)**	**M**	**SD**
Pain	4.31	3.12
Fatigue	5.79	2.64
Nausea	0.86	1.88
Depression	4.06	3.23
Anxiety	3.47	3.24
Drowsiness	3.90	3.10
Dyspnea	1.85	2.93
Loss of appetite	3.76	3.24
Reduced wellbeing	5.43	2.58
Sleep disorders	3.35	2.98
**Classification of symptoms**	**M**	**SD**
Physical (range 0 to 60, 60 being the worst possible severity)	20.61	10.39
Emotional (range 0 to 40, 40 being the worst possible severity)	7.53	5.94
Total ESAS symptoms (range 0 to 100, 100 being the worst possible severity)	32.25	15.69

ESAS: Edmonton Symptom Assessment System, M: Mean, SD: Standard Deviation.

**Table 3 healthcare-11-01272-t003:** Self-reported quality of life assessed with the EORTC QLQ-C 30 in 72 cancer patients.

Scores from 0 to 100	M	SD
Global health status (100 indicates the best condition)	46.30	23.27
Functional scales (100 indicates the best condition)		
Physical functioning	60.65	32.80
Role functioning	61.57	34.22
Emotional functioning	39.51	25.80
Cognitive functioning	28.70	27.72
Social functioning	52.78	32.26
Symptom scales (0 indicates the best condition)		
Fatigue	52.16	24.27
Nausea/vomiting	5.56	16.55
Pain	45.60	33.10
Dyspnea	23.61	32.82
Insomnia	35.65	34.18
Loss of appetite	27.78	34.03
Constipation	9.72	22.68
Diarrhoea	7.45	21.10
Financial difficulties	17.50	26.14

EORTC QLQ-C 30: European Organization for Research and Treatment of Cancer Quality of Life Questionnaire-Core 30 version 3, M: Mean, SD: Standard Deviation.

**Table 4 healthcare-11-01272-t004:** Results of the Client Satisfaction Questionnaire-8 in 72 in-home palliative cancer patients.

Items (Range 1 to 4, Where 4 Is the Best Condition)	M	SD
How would you rate the quality of service you received?	3.18	0.64
Did you get the kind of service you wanted?	2.06	0.71
To what extent has our service met your needs?	2.24	0.90
If a friend were in need of similar help, would you recommend our service to him or her?	2.00	0.75
How satisfied are you with the amount of help you received?	1.86	0.92
Have the services you received helped you to deal more effectively with your problems?	2.88	0.71
In an overall, general sense, how satisfied are you with the service you received?	3.43	0.67
If you were to seek help again, would you come back to our service?	2.08	0.85
**Total score of the CSQ-8 (range 8 to 32, where 32 is the best condition)**	19.72	3.34

CSQ-8: Client Satisfaction Questionnaire-8, M: Mean, SD: Standard Deviation.

**Table 5 healthcare-11-01272-t005:** Bivariate associations of satisfaction with health care, self-reported quality of life, and symptoms of 72 palliative cancer patients.

Spearman’s Rho	CSQ-8	ESAS	EORTC QLQ-C30
**CSQ-8**	-	−0.397 *	0.486 *
**ESAS**	−0.296 *	-	−0.490 *
**EORTC QLQ-C30**	0.426 *	−0.490 *	-

* means *p* < 0.05. CSQ-8: Client Satisfaction Questionnaire-8, ESAS: Edmonton Symptom Assessment System, EORTC QLQ-C30: European Organisation for Research and Treatment of Cancer Quality of Life Questionnaire-Core 30 version 3.

**Table 6 healthcare-11-01272-t006:** Regression analysis of the CSQ-8 with quality of life and symptoms of 72 cancer patients.

	B	SE	*p*
Edmonton Symptom Assessment System	−0.504	0.017	0.001
EORTC QLQ-C30	−0.640	0.026	0.002

Dependent variable: CSQ-8, R^2^ = 0.338, with *p* < 0.001. CSQ-8: Client Satisfaction Questionnaire-8, EORTC QLQ-C30: European Organisation for Research and Treatment of Cancer Quality of Life Questionnaire-Core 30 version 3.

## Data Availability

Study data are available on reasonable request to the corresponding author.

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
