# Peer review of "Cancer Patients’ Satisfaction with In-Home Palliative Care and Its Impact on Disease Symptoms"

_healthcare, 2023, doi:10.3390/healthcare11091272_

Round 1

Reviewer 1 Report

Interresing analysis. 

I suggest to give more results and to discuss more deeply the results

Majors coments

1. Can the authors report if patients received systemic treatment, and if no, for how long the systemic treatement were ended

2. The authors discuss the mean duration of palliative care in the discussion section, but I was not able to found this result in the result section.

3. The results of CSQ68 should be reported by items and not s a global note. The results of this assesment should be discussed more deeply in th discussion section 

4. The table are diffiult to read. I suggest to reporte the range and what mean the limit of each table

5. on the results, section, I suggest not only to report the number of correcation but also what these correlations mean (ie a worde quality f life is associated with....)

6. In the discussion section, the authors reported that "in multivariate analysis...." It's confusing because multivariate analysis seems not reported on method section

Author Response

Point 1: Interresing analysis. I suggest to give more results and to discuss more deeply the results.

Response 1: Thank you for your feedback. We have revised the manuscript to include more detailed results and a deeper discussion of our findings. We hope these changes will address your concerns and improve the quality of the manuscript.

Point 2: Can the authors report if patients received systemic treatment, and if no, for how long the systemic treatement were ended

Response 2: All individuals included in the study had previously received treatment for cancer through surgery, radiotherapy, or chemotherapy. At the time of the study, none of the patients were receiving systemic treatment, and the treatment provided was focused on symptom improvement. However, data on the duration of time since the suspension of systemic treatment was not collected. This information has been detailed in the manuscript (Methods and Limitations). We are grateful for the time the reviewer has invested in improving our article.

Point 3: The authors discuss the mean duration of palliative care in the discussion section, but I was not able to found this result in the result section.

Response 3: It is provided in Table 1. We have trying to better present it in results and discussion. Thank you for your comments.

Point 4: The results of CSQ68 should be reported by items and not s a global note. The results of this assesment should be discussed more deeply in th discussion section

Response 4: We have revised the Results section to report the CSQ8 results by items, as suggested. We have also tried to provide a more detailed discussion of the CSQ8 results in the Discussion section, exploring the implications of the findings for the quality of care received by patients in our study. We hope these changes address your concerns and improve the overall quality of our manuscript. 

Point 5: The table are diffiult to read. I suggest to reporte the range and what mean the limit of each table

Response 5: All tables have been revised according the suggestion. Thank you.

Point 6: on the results, section, I suggest not only to report the number of correcation but also what these correlations mean (ie a worde quality f life is associated with....)

Response 6: We agree. The results section has been revised after suggestion in the hope that it is clearly presented now.

Point 7: In the discussion section, the authors reported that "in multivariate analysis...." It's confusing because multivariate analysis seems not reported on method section

Response 7: We apologize again for any confusion that may have arisen from our manuscript. We have reviewed our methods section with the multivariate analyses, in the statistical analysis subsection. We have revised the manuscript to clarify this point. The results section has been also better detailed according to this comment and the aforementioned. Thank you very much for your valuable revision.

Reviewer 2 Report

This manuscript reports on questionnaire data on quality of life, symptom severity and satisfaction with care received obtained from 72 patients receiving in-home palliative care. These are interesting data worth publishing.

The following comments may be used to improve the manuscript.

1. The Abstract and Results section 3.3. refer to a ‘QlG-8’ score. This score is not described in the Methods section and this may be a spelling error.

2. When presenting associations among ESAS, QLQ-C30 and CSQ-8, I would prefer to read ‘negative association’ when referring to a negative correlation, i.e. when r<0.

3. Abstract, Discussion and Conclusion state that assessment of patient satisfaction should be included in routine care. I agree with the authors that the described associations of symptom load, quality of life and patient satisfaction are ‘obvious’ to health care professionals in palliative care (Discussion, para 7).  I also agree that an empirical demonstration of these associations is worthwhile and helpful. However, I am not convinced that routine assessment of patient satisfaction will improve care in addition to essential efforts to reduce symptom severity. I recommend to improve the arguments presented to support the authors’ conclusion.

I encourage to engage a native English speaker or high-quality software to check the style.

Author Response

This manuscript reports on questionnaire data on quality of life, symptom severity and satisfaction with care received obtained from 72 patients receiving in-home palliative care. These are interesting data worth publishing.

The following comments may be used to improve the manuscript.

 Point 1: The Abstract and Results section 3.3. refer to a ‘QlG-8’ score. This score is not described in the Methods section and this may be a spelling error.

Response 1: Thank you for your valuable feedback. We apologize for any confusion caused by the mention of a 'QlG-8' score in the Abstract and Results section. This was indeed a typographical error and should have been referred to as 'CSQ-8'. We have now corrected this error in the revised version of the manuscript.

Point 2: When presenting associations among ESAS, QLQ-C30 and CSQ-8, I would prefer to read ‘negative association’ when referring to a negative correlation, i.e. when r<0.

Response 2: We agree with the reviewer. Moreover, we have revised all the manuscript in the hope that is clearly written in the current version. Thank you very much.

Point 3: Abstract, Discussion and Conclusion state that assessment of patient satisfaction should be included in routine care. I agree with the authors that the described associations of symptom load, quality of life and patient satisfaction are ‘obvious’ to health care professionals in palliative care (Discussion, para 7).  I also agree that an empirical demonstration of these associations is worthwhile and helpful. However, I am not convinced that routine assessment of patient satisfaction will improve care in addition to essential efforts to reduce symptom severity. I recommend to improve the arguments presented to support the authors’ conclusion.

Response 3: Thank you for your feedback on our manuscript. We have carefully considered your points and have revised the abstract/discussion/conclusion accordingly. We understand your concerns about routine assessment of patient satisfaction and its impact on care improvement. We have trying to further strengthen our arguments to support this conclusion in the revised version of the manuscript. We hope you find these changes satisfactory, and thank you again for your valuable input.

Comments on the Quality of English Language: I encourage to engage a native English speaker or high-quality software to check the style.

Response: We have revised the manuscript and made improvements to the language and style to the best of our abilities by applying a high-quality software. If necessary, after final acceptance of the article, we will consider engaging a professional translator to ensure the highest quality language. Thanks.

Reviewer 3 Report

The paper presents a cross-sectional study, conducted in the primary health care sector in six clinical management units with the aim to determine whether the satisfaction of cancer patients with in-home palliative care is associated with the impact of disease symptoms and with self-perceived quality of life. The instruments used were: the severity of disease symptoms was determined by the Edmonton Symptom Assessment System (ESAS); the quality of life was evaluated with the EORTC QLQ-C30 questionnaire, and the patient’s satisfaction with the care received, by the CSQ-8. The results were based on correlational analysis and linear regression.

The research and the findings are useful, however, there are some deficiencies in the methodological section. I propose a minor revision to add and provide the following:

-Reliability measurements for the scales used, e.g. Cronbach alpha.

-The correlation matrix that facilitates the reader to see all correlations together.

-A Table presenting the results of the linear regression, with all related information.

-Present the limitations of the study in a separate subsection.

The paper is readable but some proofreading will improve it

Author Response

The paper presents a cross-sectional study, conducted in the primary health care sector in six clinical management units with the aim to determine whether the satisfaction of cancer patients with in-home palliative care is associated with the impact of disease symptoms and with self-perceived quality of life. The instruments used were: the severity of disease symptoms was determined by the Edmonton Symptom Assessment System (ESAS); the quality of life was evaluated with the EORTC QLQ-C30 questionnaire, and the patient’s satisfaction with the care received, by the CSQ-8. The results were based on correlational analysis and linear regression.

The research and the findings are useful, however, there are some deficiencies in the methodological section. I propose a minor revision to add and provide the following:

Point 1: Reliability measurements for the scales used, e.g. Cronbach alpha.

Response 1: Thank you for your comment. The psychometric properties for the scales used in the study have been detailed in the Method section with their corresponding references.

Point 2: The correlation matrix that facilitates the reader to see all correlations together.

Response 2: We agree that presenting the correlation matrix is a useful way to visualize the relationships between variables. We have revised the Results section to include the correlation matrix in order to provide a clearer and more comprehensive presentation of the data according the suggestion.

Point 3: A Table presenting the results of the linear regression, with all related information.

Response 3: It has been included after suggestion. Thank you very much for your appreciated comments.

Point 4: Present the limitations of the study in a separate subsection.

Response 4: Corrected after suggestion.

Reviewer 4 Report

Dear editor and dear authors,

Thank you for the opportunity to review your paper entitled “Cancer patients’ satisfaction with in-home palliative care and its impact on disease symptoms.”

 The abstract was well written. However, we suggest presenting the main results in the abstract.

The introduction, results, and discussion sections are well-framed.

In the material and methods section, the authors considered that the “type of cancer and duration of palliative care” is sociodemographic data. Still, we assumed that are clinical data (characteristics). The authors said that in the results.

Limitations are identified. We suggested improving the conclusions. The authors should explain to readers how these results can be applied in practice and what we should change after this research. In the manuscript, the aim of the study was “to determine whether satisfaction with the health care received is associated with the severity of symptoms and with self-perceived quality of life, for patients with advanced cancer receiving in-home palliative care”, the authors should say in conclusion if they achieve the aim or not.

Author Response

Dear editor and dear authors,

Thank you for the opportunity to review your paper entitled “Cancer patients’ satisfaction with in-home palliative care and its impact on disease symptoms.”

Point 1: The abstract was well written. However, we suggest presenting the main results in the abstract.

Response 1: We agree with the reviewer. The abstract has been rewritten and the main results have been better detailed in the hope that it presents now a better summary of our study. All changes have been highlighted in yellow. Thank you very much for your comments.

Point 2: The introduction, results, and discussion sections are well-framed. In the material and methods section, the authors considered that the “type of cancer and duration of palliative care” is sociodemographic data. Still, we assumed that are clinical data (characteristics). The authors said that in the results.

Response 2: Thank you for your comments. We have taken them into consideration and made the necessary changes which have been highlighted in yellow. In the materials and methods section, we have relocated the information regarding the "type of cancer and duration of palliative care" to the section on clinical characteristics, as suggested. We have also maintained our clarification on this matter in the results section. Thank you again for your valuable feedback.

Point 3: Limitations are identified. We suggested improving the conclusions. The authors should explain to readers how these results can be applied in practice and what we should change after this research. In the manuscript, the aim of the study was “to determine whether satisfaction with the health care received is associated with the severity of symptoms and with self-perceived quality of life, for patients with advanced cancer receiving in-home palliative care”, the authors should say in conclusion if they achieve the aim or not.

Response 3: Thank you for your thoughtful review of this section. We have carefully considered your comments and have made the necessary revisions to the conclusion to address your concerns. We have provided a clear explanation of how the results of our study can be applied in practice for patients with advanced cancer receiving in-home palliative care. Furthermore, we have explicitly stated whether we achieved the aim of the study. We appreciate your valuable feedback and believe that the revised conclusion section has been improved and better presents our findings and applications. Thank you again for your time and consideration.

Reviewer 5 Report

The aim of the study was to determine whether the satisfaction of cancer patients with the in-home palliative care is associated with the impact of disease symptoms and with self-perceived quality of life.
The topic taken up is extremely topical, as in recent years an increase in the importance of home palliative care can be observed. The study was planned and conducted properly. Three standardized research tools were used for this purpose. The study group is large enough to draw reliable conclusions. The research project was carried out in accordance with ethical standards after prior approval by the ethics committee (consent number attached).
A certain insufficiency is caused by the paragraph containing conclusions. It's very short. It consists of two sentences. The authors should expand on this part.

Author Response

The aim of the study was to determine whether the satisfaction of cancer patients with the in-home palliative care is associated with the impact of disease symptoms and with self-perceived quality of life.

The topic taken up is extremely topical, as in recent years an increase in the importance of home palliative care can be observed. The study was planned and conducted properly. Three standardized research tools were used for this purpose. The study group is large enough to draw reliable conclusions. The research project was carried out in accordance with ethical standards after prior approval by the ethics committee (consent number attached).

Point 1: A certain insufficiency is caused by the paragraph containing conclusions. It's very short. It consists of two sentences. The authors should expand on this part.

Response 1: Thank you very much for your comments. Regarding the conclusions, this part has been rewritten in the hope that it is more completed and suitable for publication. We agree with the reviewer. All changes have been highlighted in yellow.

Round 2

Reviewer 1 Report

no more comments, thanks for the responses